# Disease progression role as well as the diagnostic and prognostic value of microRNA-21 in patients with cervical cancer: A systematic review and meta-analysis

**Alemu Gebrie** [ORCID]*

Department of Biomedical Sciences, School of Medicine, Debre Markos University, Debre Markos, Ethiopia

* alemugebrie2@gmail.com

**Data Availability Statement:** All relevant data are within the manuscript and its Supporting Information files.

## Abstract

### Introduction

Cervical cancer is the fourth commonest and the fourth leading cause of cancer death in females globally. The upregulated expression of microRNA-21 in cervical cancer has been investigated in numerous studies, yet given the inconsistency on some of the findings, a systematic review and meta-analysis is needed. Therefore, the aim of this systematic review and meta-analysis is to investigate the role in disease progression as well as the diagnostic and prognostic value of microRNA-21 in patients with cervical cancer.

### Methods

Literature search was carried out through visiting several electronic databases including PubMed/MEDLINE/ PubMed Central, Web of Science, Embase, WorldCat, DOAJ, Science-Direct, and Google Scholar. After extraction, data analysis was carried out using Rev-Man 5.3, STATA 15.0 and Meta-disk 1.4. I2 and meta-bias statistics assessed heterogeneity and publication bias of the included studies, respectively. The area under summary receiver operating characteristic curve and other diagnostic indexes were used to estimate diagnostic accuracy.

### Result

A total of 53 studies were included for this systematic review and meta-analysis. This study summarized that microRNA-21 targets the expression of numerous genes that regulate their subsequent downstream signaling pathways which promote cervical carcinogenesis. The targets addressed in this study included TNF-α, CCL20, PTEN RasA1, TIMP3, PDCD-4, TPM-1, FASL, BTG-2, GAS-5, and VHL. In addition, the meta-analysis of reports from 6 eligible studies has demonstrated that the overall area under the curve (AUC) of summary receiver operating characteristic (SROC) of microRNA-21 as a diagnostic accuracy index for cervical cancer was 0.80 (95% CI: 0.75, 0.86). In addition, evidence from studies revealed that upregulated microRNA-21 led to worsening progression and poor prognosis in cervical cancer patients.

**Funding:** The author(s) received no specific funding for this work.

**Competing interests:** The authors have declared that no competing interests exist.

## Conclusion

microRNA-21 is an oncogenic microRNA molecule playing a key role in the development and progression of cervical malignancy. It has good diagnostic accuracy in the diagnosis of cervical cancer. In addition, the upregulation of microRNA-21 could predict a worse outcome in terms of prognosis in cervical cancer patients.

## Introduction

Cervical cancer, which develops in a females' cervix, is the fourth commonest malignancy (next to breast, colorectal, and lung cancer) and the fourth leading cause of cancer death in females globally. Nearly all cervical cancer cases (99.7%) are principally attributable to the widespread infection with high-risk strains of an extremely common sexually transmitted oncovirus called human papillomavirus (HPV) [1–4]. About 70% of cervical carcinoma cases and pre-cancerous cervical lesions are specifically linked with HPV 16 and HPV 18 strains in all countries [5]. Sexual contact is the primary HPV mode of transmission and most people are infected shortly after the onset of sexual activity [6]. In addition, quite a lot of studies have demonstrated that, low economic status, poor personal and sexual hygiene, early onset of coitarche, multiple sexual partners, high-risk sexual partner, history of sexually transmitted infections, history of vulval or vaginal precancerous and cancerous lesions, smoking, oral contraceptive pills, and immunocompromise are the cofactors for cervical cancer [7–14]. In addition, microRNAs play a role in cervical carcinogenesis as well [15].

Chronic infection with the virus results in cervical cancer in women though the majority of infections with the virus cause no symptoms and resolve spontaneously within 1–2 years [2, 5, 16]. Worldwide, it is estimated that 604,127 women were diagnosed with cervical malignancy and 341,831 women died of the cancer in 2020 alone with a geographic variation in morbidity and mortality rates. Low and middle-income countries account about 85% of worldwide deaths from this cancer [3, 17]. Also, the age-standardized incidence rate of cervical cancer is about 13.1 per 100 000 women and varies among countries from less than 2 to 75 per 100, 000 women [18].

Effective HPV vaccination (primary) and screening for, and treating precancerous lesions (secondary prevention approaches) can prevent the majority of cervical cancer cases [18]. Luckily, cervical cancer is one of the most successfully treatable forms of cancer when it is early detected and effectively managed. It can be cured if diagnosed at an early stage and treated on time. In addition, cervical cancer diagnosed in late stages can also be managed with appropriate treatment and palliative care. Therefore, given a comprehensive approach to prevent, screen and treat, cervical malignancy can possibly be extirpated as a public health problem within a generation [2]. In 2020, World Health Organization (WHO) adopted a global strategy for eliminating cervical cancer through a triple pillar intervention strategy: 90% of girls fully vaccinated by the age of 15 years, 70% of women screened by the age of 35 years and again by 45 years, and 90% of women with precancer treated and 90% of women with invasive cancer managed [19].

The progression of HPV infected epithelial cells to invasive cervical cancer is a long-standing process accompanied by the accumulation of DNA changes in host cell genome. The changes include both epigenetic and genetic alterations in tumor suppressor genes and oncogenes [20]. To initiate the infection, the virus enters the host basal squamous cells through a micro-wound or abrasion and integrate into the host cell, where a series of genetic events

occur within the basal epithelial cervical cells thus enabling viral replication. Meanwhile, these events create a favorable environment for neoplastic progression [21]. Additionally, to ensure continuous replication in the basal epithelial cells, the virus evades the host immune system. In a stage called CIN (cervical epithelial neoplasia) 1, the cervical cells are chronically infected by the virus. At least 10–12 years are required to advance from precancerous lesion to invasive carcinoma [22, 23]. Within 2–3 years after infection, CIN 1 lesions that do not revert may advance into CIN 2/3 [24].

The microRNAs, endogenous short (17–22 nucleotides), highly conserved, and non-coding RNAs, are well-known for their involvement during apoptosis, gene expression, cell cycle regulation, and metastasis [25]. Like other RNAs, they are expressed from DNA yet do not code any proteins, and circulate in the blood embedded in glandular vesicles and exosomes [26]. The microRNAs, epigenetic regulators, bind to the sequence motifs within 3'-untranslated regions (3'-UTR) of targeted mRNA molecules to increase degradation or translational inhibition. Plethora of evidence demonstrated that many aberrated miRNAs target multiple tumor suppressor genes and oncogenes with multiple mechanisms playing a very complex role in cervical carcinogenesis development and progression [27–32].

Specifically, studies revealed that microRNA-21 (5'UAGCUUAUCAGACUGAUGUUGA3') activates cell proliferation in HeLa cells and its inhibition suppressed cell multiplication by increased expression of tumor suppressor gene called programmed cell death 4 (PDCD4), an apoptotic protein, indicating that microRNA-21 is a key oncogenic molecule that is upregulated in cervical cancer [28, 33]. The gene for human microRNA-21 is located in the loci (fragile site, FRA17B) within the 17q23.2 chromosomal region [34] where the human HPV integrates. HPV integration into the human genome results in epigenetic and genetic changes, justifying that the proximity of the gene for microRNA-21 and the site for the integration of HPV could be imputed to its upregulation in cervical cancer [27, 32, 35]. In addition, changes in the level of microRNA-21 have been shown to affect the expression of proteins working at certain steps in key signaling pathways such as caspase-8 and caspase-3 in apoptosis, and an important tumor suppressor protein known as phosphatase and tensin homolog (PTEN) [36, 37]. Furthermore, upregulated expression of microRNA-21 has been shown in various studies conducted using the whole blood of the patients and cervical cancer cell lines, including SiHa, HT-3 and HeLa cells [36–42].

Because miRNAs are highly stable in body fluids, circulating miRNAs are ideal molecules as biomarkers for the diagnosis and prognosis of cervical malignancy [43]. The level of microRNA-21 is significantly upregulated in cervical cancer patients, indicating that it could serve as a diagnostic and prognostic biomarker for cervical cancer [44, 45]. The upregulated expression of microRNA-21 in cervical cancer and its importance as a diagnostic and/or prognostic role have been investigated in numerous studies, yet given the inconsistency on some of the findings, a systematic review and meta-analysis is needed. Besides, there is a growing need and realistic potential for non-invasive diagnostic and prognostic molecular cancer biomarkers for early detection, and monitoring the recurrence of cervical cancer [46, 47]. Therefore, the aim of this systematic review and meta-analysis is to investigate the role in disease progression as well as the diagnostic and prognostic value of microRNA-21 in patients with cervical cancer.

## Materials and methods

### Study design and literature searching strategies

A systematic review and meta-analysis was carried out to compile the most contemporary evidence using published studies as well as grey literatures on disease progression role as well as the diagnostic and prognostic value of microRNA-21 in patients with cervical cancer. For the

design, conducting and reporting as well as rigor of this study, the updated protocol of the Preferred Reporting Items for Systematic Reviews and Meta-Analyses (PRISMA) 2020 guideline has been followed [48] (**S1 Table**). This review was also carried out as per the guideline recommended by the Human Genome Epidemiology Network for systematic review of genetic-association studies [49]. To search potentially relevant studies, two strategies were followed. These are, the basic and advanced electronic database searching at PubMed/MEDLINE/ PubMed Central, Web of Science, Embase, WorldCat, DOAJ, ScienceDirect, and Google Scholar as well as the manual search of the lists of references of related studies to increase the searching sensitivity and retrieve more eligible articles.

"tissue/serum/plasma/circulating/urine 'microRNA-21', 'miRNA-21', 'microRNA 21', 'miR-21', 'miRNA 21', 'miR-21-5p", miR-21-3p, cervical cancer/cervical neoplasm/cervical tumor", cervical cancer/neoplasm/tumor prognosis", "cervical cancer/neoplasm/tumor diagnosis", "cervical cancer disease progression", "signaling pathways", and "targets" were the key-terms used for searching the articles in the reputable databases taken, as appropriate, both in separation and in all possible combination using the Boolean operator: "OR", "AND", and "NOT". In addition, the PubMed database have been searched by using Medical Subject Headings (MeSH) terms. Exhaustive searching of studies has been conducted from November 26, 2021 to January 18, 2022 and all the records accessed until January 18, 2022 were deemed eligible in the selection process. The searching has been carried out by two independent persons (AG and GM) in accordance with the PICO acronym (participants, interventions/index tests, comparisons, outcomes/target conditions). Furthermore, EndNote version 20 reference manager has been used for handling references in the study.

## Eligibility criteria

**Inclusion criteria.** The contents of each retrieved study have been scrupulously reviewed by using a preset criterion. After that, the eligible articles fulfilling the following criteria were finally included in the present review as seen in **Table 1**.

*Population.* Articles conducted among cervical cancer patients were considered. All patients identified as cervical cancer cases in the articles were diagnosed with pathological diagnosis, the gold standard test.

*Study area.* Studies which have been conducted anywhere in the world were considered in the study.

*Study design.* Original articles that contain data reporting disease progression role as well as the diagnostic and prognostic value of microRNA-21 in patients with cervical cancer were considered.

*Language.* Studies which have been published only in English language were considered.

*Publication condition.* Articles that fulfill the eligibility criteria were included regardless of their publication status (published, unpublished and grey literature, etc.)

**Table 1. Eligibility criteria using PICO (participants, interventions/index test, comparisons, outcomes/target condition) criteria.**

| Component | Eligibility criteria |
|---|---|
| Participant | Cervical cancer OR cervical neoplasm OR cervical tumor patients |
| Index test | microRNA-21 OR miRNA-21 OR microRNA 21 OR miR-21 OR miRNA 21 OR miR-21-5p OR miR-21-3p |
| Comparison | Healthy control participants |
| Outcome/target condition | Disease progression OR Diagnosis OR Prognosis in the subjects |

*Study sample type.* Those studies detecting microRNA-21 expression in tissues, plasma, serum or any other human body fluids were included in the study.

**Exclusion criteria.** Those studies in which microRNA-21 expression were carried out with no survival analysis or prognosis parameters were excluded. Articles with a combined data or report of microRNA molecules instead of a single microRNA-21 were excluded. Reviews, commentaries, letters, conference abstracts, case reports, laboratory or animal studies were also excluded from the meta-analysis. Furthermore, inadequate data reported for conducting the meta-analysis were excluded. In other words, articles with missing of important data such area under the cure, sensitivity, and specificity of diagnostic accuracy data with respective confidence intervals were excluded from being eligible for the meta-analysis.

## Data abstraction

Two evaluators (AG and MS) separately examined and extracted the necessary data using a pre-set data extraction tool and any discrepancies among the reviewers were resolved by discussion. The following data related to diagnostic and prognostic values from each article were extracted: first author, year of publication, country, cancer stage, sample source, microRNA-21 assay method, and cut-off value for microRNA-21 expression to stratify low and high subject groups, sensitivity, specificity, true positive, false positive, false negative, true negative, likelihood ration, area under the cure receiver operating characteristics. There were attempts to contact, twice, the corresponding authors of the articles in which there were incomplete data to be extracted and included in the meta-analysis, but none of them responded and those studies were excluded from the meta-analysis.

## Operational definition/interpretation of outcome variable

The AUROC (Area under receiver operating characteristics curve) is defined or interpreted as the probability that a randomly chosen diseased subject is ranked or rated as more likely to be in disease state than a randomly chosen non-diseased one. The definition/interpretation is according to nonparametric Mann-Whitney U statistics which is unitized in calculating the AUC (area under the curve). In other words, it is the mean value of sensitivity for all the possible values of specificity [50, 51].

## Article quality evaluation

For diagnostic component, the reviewers (AG and MS) used an evidence-based quality assessment tool which was developed for use in systematic reviews of diagnostic accuracy studies-2 (QUADAS-2) to evaluate the quality of each study [52]. This tool has 4 domains (patient selection, index test, reference standard, and flow and timing). All domains are assessed in context of "risk of bias", and the first 3 domains are evaluated in terms of "concerns about applicability". Signaling questions are also included for judging risk of bias. The signaling questions are answered "yes", "no", or "unclear". There is also a description box to help answer "yes", "no", or "unclear" to be more explicit. The quality assessment is rated, both for risk of bias and applicability concerns, as "High", "Low", and "Unclear".

For prognostic component, the New Castle Ottawa Scale (NOS) was used to assess the quality of all included research. The NOS consists of eight components that are divided into three categories (selection, comparability, and outcome or exposure). To measure the quality of the included studies, a star system is used, with the highest quality research receiving a maximum of one star for an item, except for the item relating to comparability, which receives two stars. The NOS scale runs from zero to nine stars, with nine being the best quality study. Each study that received more than five stars was deemed to be of high quality.

## Statistical analysis

Different statistical analyses in this meta-analysis of diagnostic importance of microRNA-21 in cervical cancer patients were performed using MetaDiSc 1.4 (Cochrane Colloquium, Barcelona, Spain), MedCalc- version 20.023, Review Manager 5.3, and Stata 11.0 (StataCorp, College Station, TX, USA) software. The statistical analyses were 2-sided and a value of <0.05 was considered statistically significant.

To analyze the diagnostic accuracy of microRNA-21 in cervical cancer, the summary diagnostic indexes: sensitivity and specificity, positive likelihood ratio (PLR), negative likelihood ratio (NLR), and diagnostic odds ratio (DOR) with the corresponding 95% CIs were calculated. The indexes with significant heterogeneity were analyzed using the random-effects model (DerSimonian-Laird method) whereas those with less heterogeneity were analyzed using fixed-effects model (Mantel–Haenszel method). In addition, the summary receiver operating characteristic (SROC) curve was carried out to determine the overall diagnostic accuracy in the different threshold points. Subgroup analysis was carried out by dividing the studies according to different sample types. Heterogeneity among studies was assessed using Cochran's Q test and the $I^2$ statistic. values <25%, 25%-50%, and >50% were set to indicate mild, moderate, and significant heterogeneity, respectively. A p-value<0.1 or I2 > 50% indicated the existence of significant heterogeneity [53–56]. Besides, publication bias was assessed using Egger's test and Begg's test objectively as well as the funnel plot subjectively. In addition, sensitivity analysis was carried out to assess the stability of the studies.

## Result

### Selection of eligible studies

Using the searching method, a total of 1,362 records were initially retrieved through database searching. After the removal of duplicates, the titles and abstracts for 828 records were screened for eligibility. Among them, 61 records were identified and full-text articles were accessed after retrieval, 4 studies [57–60] being not retrieved. Then, 8 articles [44, 61–67], 4 [64, 66–68] because of irrelevant index test and/or target condition, 2 [44, 61] due to incomplete data for meta-analysis, and 2 [63, 65] as the studies reported combined index test report, were excluded through assessment of the full-text articles based on the pre-set exclusion criteria, Finally, 53 studies were deemed eligible in this study. From the included 53 studies, 7 studies were included for the meta-analysis of diagnostic importance component of the study (**Fig 1**) (**S2 Table**).

### Description of the included studies

The characteristics of each included study are described in detail in **Table 2**. The main data were extracted from the studies including the first author, the country where the research was conducted, the publication year, study design, the sample source, sample size, cancer stage, cut-off value for microRNA-21 expression, assay method, and the necessary diagnostic indexes. A total of 9 studies with 10 reports of the diagnostic importance of microRNA-21 in cervical cancer were included in this study. These 9 studies included 1624 study subjects (957 cervical cancer patients and 667 healthy controls). All cervical cancer cases were diagnosed by histopathology, which is the gold standard for the diagnosis of cervical cancer. All of the included studies reporting on the diagnostic importance of microRNA-21 in cervical cancer were conducted in Asian countries: China, India, Iran, and South Korea. In addition, the publication years of the included studies were from 2015 to 2021, and almost all the studies were case-control by study design. In the studies, the expression of microRNA-21 was assayed by

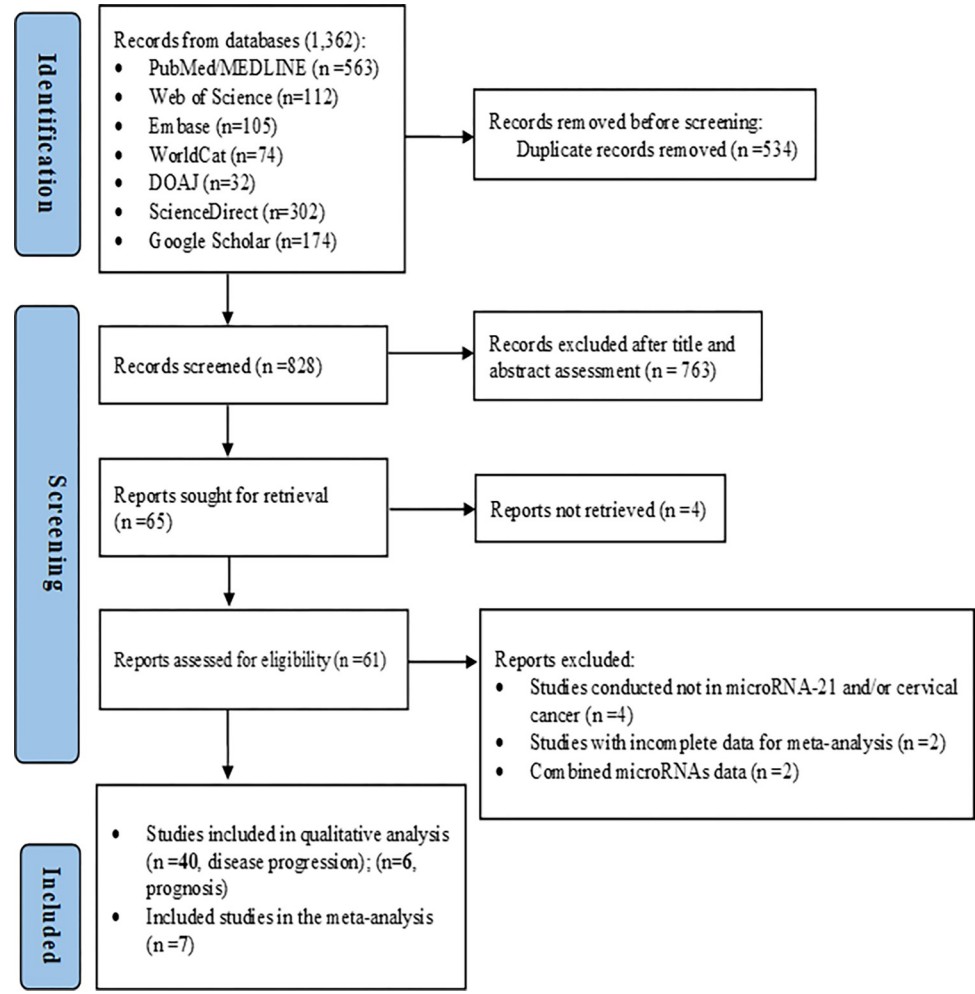

**Fig 1. PRISMA 2020 flow diagram summarising the selection process of studies in the systematic review and meta-analysis.**

quantitative reverse transcription polymerase chain reaction (RT-PCR). Tissue biopsy, serum, plasma, and urine were the specimen sources of the studies. In addition, the cancer stage, and the cut-off values of microRNA-21 were different among the included studies.

In addition, the characteristics of studies included for the disease progression and prognostic component of this study have been indicated in **S3 Table**. In the studies included for the qualitative review of disease progression role of microRNA-21 in cervical cancer, RT-PCR was used as a method of analysis for the expression of microRNA-21 in almost all of the studies. Besides, most of the included studies have been published in Q1 journals as per Scimgo journal ranking.

## Quality evaluation results

Using QUADAS-2 as a standard quality assessment tool, the quality of included studies in the meta-analysis of the present study was evaluated using Review Manager 5.3 software as depicted in **Fig 2A and 2B**. The quality of each diagnostic accuracy studies was assessed based on the QUADAS-2 tool [52] which has 4 key components: patient selection, index test, reference standard, flow and timing, and judges risk of bias and applicability concerns. While each

**Table 2. The characteristics of included studies (reports) in the systematic review and meta-analysis.**

| Author | Publication year | Country | Study design | Sample type | Cervical cancer cases | Healthy controls | Cancer stage | Cut-off value | Test method | AUROC |
|---|---|---|---|---|---|---|---|---|---|---|
| Qiu et al [44] | 2020 | China | Case-control | Serum | 112 | 90 | stage I-IIA | >2 | qRT-PCR | 0.783* |
| Ma et al [69] | 2019 | China | Case-control | Plasma | 97 | 87 | stage I-II | NR | qRT-PCR | 0.677 (0.577–0.776) |
| Zamani et al [70] | 2020 | Iran | Case-control | Tissue | 50 | 46 | NR | 0.0003284 | qRT-PCR | 0.856 (0.774–0.938) |
| Jia et al [71] | 2015 | China | Case-control | Serum | 123 | 94 | Up to stage III | NR | qRT-PCR | 0.819 (0.762-0.876) |
| Ruan et al [72] | 2020 | China | Case-control | Serum | 68 | 57 | NR | 3.855 | qRT-PCR | 0.723 (0.631–0.815) |
| Du et al [61] | 2020 | China | Case-control | Serum | 140 | 140 | stage I-II | NR | qRT-PCR | < 0.7* |
| Park et al [73] | 2017 | Korea | Case-control | Tissue | 52 | 50 | stage I-IIA | >1.975 | qRT-PCR | 0.833 (0.753–0.912) |
| Aftab et al [74] | 2021 | India | prospective | Urine | 50 | 50 | stage I-IV | 1.912 | qRT-PCR | 0.971 (0.957–0.985) |
| Aftab et al [74] | 2021 | India | prospective | Tissue | 40 | 30 | stage I-IV | 1.182 | qRT-PCR | 0.993 (0.975–1.000) |
| Zhu et al [75] | 2018 | China | Case-control | Tissue | 25 | 23 | NR | NR | qRT-PCR | 0.871 (0.743–0.950) |

NR: Not Reported

*This article was not included in the meta-analysis because only point estimate can be extracted from the article; AUROC: Area under receiver operating characteristics curve

of the components was subjectively evaluated with respect to risk of bias, only the first 3 domains were evaluated in terms of applicability concerns. The domains in both risk of bias and concerns regarding applicability were graded as "high", "unclear", or "low". The evaluation of the studies revealed that most of the components in almost all the included studies had low risk of bias and low concern of applicability.

For prognostic part, the NOSs for all relevant research were given more than five stars, indicating that the studies were of acceptable quality (**S3 Table**).

## Disease progression role of microrna-21 in cervical cancer

Plenty of evidence demonstrated that microRNA-21 is over-expressed in cervical cancer development and progression. It is functionally involved in many important hallmarks of cancer, including evading growth inhibitors, enabling proliferative immortality, activation of invasion and metastasis, apoptosis inhibition, and continual proliferative signaling [76–82]. Most of its reported direct targets are tumor suppressors which include different ligands, receptors and relay proteins of several signaling pathways with many cross-talk of proteins in the pathways [83–85]. The microRNA molecule acts by binding to the 3'-untranslated regions (3'-UTR) of targeted mRNA molecules which encode the tumor suppressors thus increasing their degradation or inhibiting their translation. The role of microRNA-21 in cervical carcinogenesis explicating the signaling pathways has been systematically summarized as it is depicted in **Fig 3**.

Although the direct target is yet to be known, research has shown that microRNA-21 indirectly upregulates TNF-α level in cervical cancer cell lines. Cellular proliferation, angiogenesis, migration, activated immune response and inhibited apoptosis result when TNF-α binds to its receptor (TNFR2) and activates nuclear factor κB (NF-κB) which upregulates c-Jun N-terminal kinase (JNK), and downregulates caspase-3 in the pathway [36, 86].

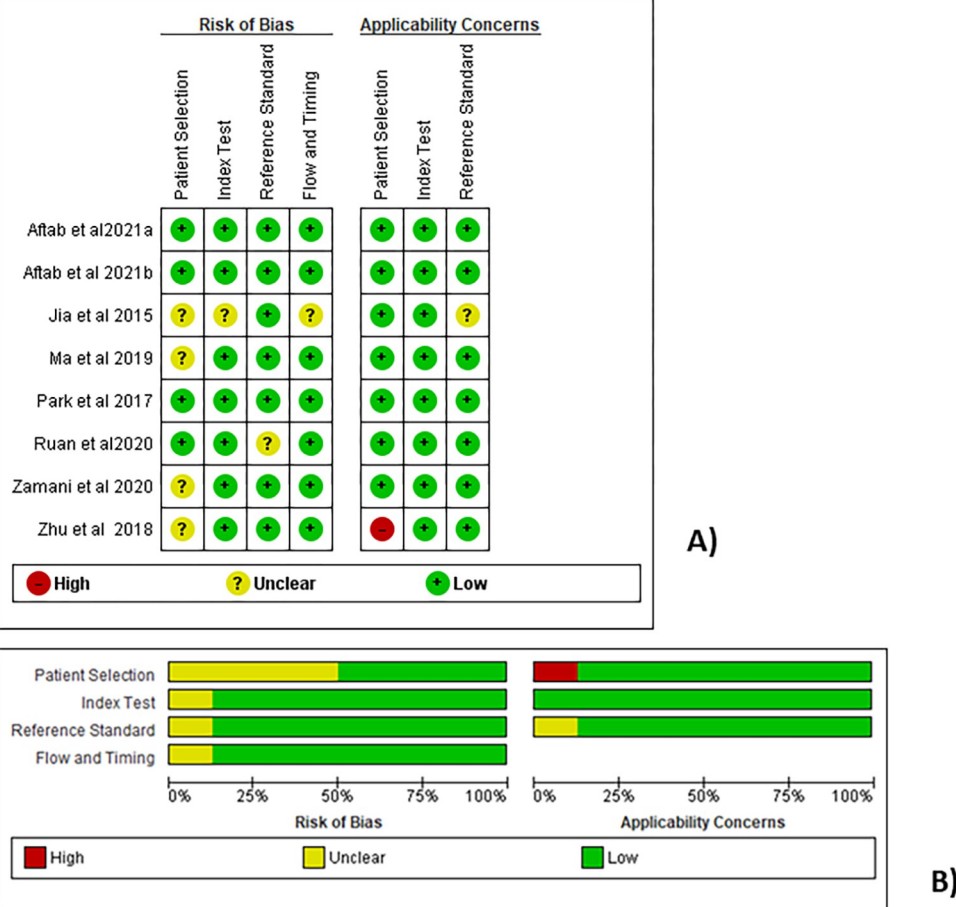

**Fig 2. The quality assessment results of included studies in the meta-analysis, carried out using RevMan 5.3 software based on the QUADAS-2 evaluation tool.** (A) Risk of bias and applicability concerns summary. Review author judgements about each domain for each study included. (B) Risk of bias and applicability concerns graph. Review author judgements about each domain are presented as percentages across the included studies.

Similarly, microRNA-21 enhances the process of cervical carcinogenesis by upregulating the PI3K/AKT/mTOR signaling pathways which ultimately induce cell proliferation, migration, growth and survival [87]. The PI3K/Akt/mTOR signaling pathways are important in various physiological and pathological conditions with cell growth and survival implications [88]. It is documented that PTEN is overexpressed significantly in cells knockout of microRNA-21 [37]. It has also been revealed that an microRNA-21 inhibitor increased G1 arrest in the cell cycle, increased cell proliferation, and induced cell apoptosis by affecting the PTEN/Akt pathway [89]. Hence, microRNA-21 works on the signaling pathways by directly targeting and downregulating PTEN that controls, in turn, the signaling pathways negatively with an overall effect of enhanced proliferation and cell survival [90].

In addition, microRNA-21 induces cell proliferation, prevents apoptosis, and activates malignant transformation by regulating RAS/RAF/MEK/ERK signaling pathway [91]. RAS is an oncogenic protein observed to be mutated in cancers thereby resulting in a constitutive stimulation of RAS/MEK/ERK pathway which activates cellular proliferation, transformation as well as anti-apoptosis signaling [92]. A member of Ras-GTPase-activating family called RasA1 which inhibits RAS has been demonstrated to be downregulated by microRNA-21 targeting the 3′-UTR of the mRNA by Luciferase activity assay [40]. Therefore, cell proliferation

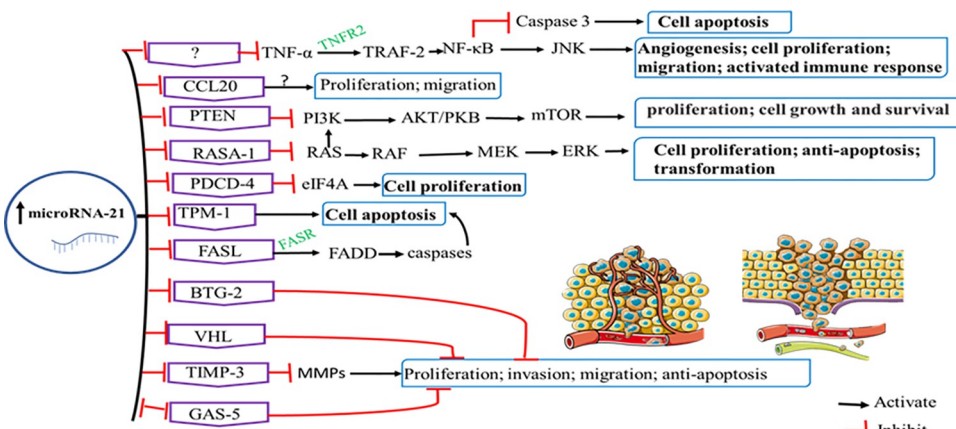

**Fig 3. The role of microRNA-21 in the cancer progression via different signaling pathways.** TNF-α: tumor necrosis factor α; TNFR: tumor necrosis factor receptor; CCL20: chemokine ligand 20; TRAF-2: TNF receptor-associated factor 2; NF-KB: nuclear factor κB; JNK: c-Jun N-terminal kinase; PTEN: phosphatase and tensin homolog; PI3K: phosphoinositide 3-kinase; AKT/PKB: Akt/protein kinase B; mTOR: mammalian target of rapamycin; RASA-1: Ras P21 Protein Activator 1; RAS: Ras protein; RAF: Raf proto-oncogene; MEK: mitogen-activated protein kinase kinase; ERK: extracellular signal-regulated kinase; PDCD-4: programmed cell death 4; eIF4A: eukaryotic initiation factor 4A, TPM-1: tropomyocin-1; FASL: Fas ligand; FASR: fas ligand receptor; FADD: Fas-associated via death domain; BTG-2: B-cell translocation gene 2; VHL: von Hippel-Lindau tumor suppressor; TIMP-3: Tissue inhibitor of metalloproteinases 3; MMPs: matrix metalloproteinases, GAS5: growth arrest-specific 5.

is induced by microRNA-21 directly targeting RasA1 and enhancing the RAS/RAF/MEK/ERK signaling pathway [93, 94]. What's more, there is a crosstalk between RAS/RAF/MEK/ERK and PI3K/AKT/mTOR signaling pathways that RAS activates the latter [95].

Cervical tumorigenesis can also be affected by microRNA-21 being a sponge of TIMP3/MMPs pathway. An aberration in the function of extracellular matrix in cancer tissue plays a role to tumor growth and metastasis [96]. Matrix metalloproteinases (MMPs) and TIMP3 positively and negatively enhance this derangement, respectively. MMP activity is inhibited by TIMP3 thereby increasing invasion and migration of malignant cells [96, 97]. It is also found that microRNA-21 directly targets TIMP3 leading to increased invasiveness, proliferation, and viability of cervical malignant cells [38].

Furthermore, microRNA-21 affects other direct target molecules including PDCD-4, TPM-1, FASL, and BTG-2 in order to stimulate its cervical carcinogenesis effect. It inhibits PDCD4 protein synthesis which decreases eIF4A hence promoting cell growth, tumor invasion, metastasis, and inhibiting cell apoptosis [42, 98, 99]. The oncomir microRNA-21 targets PDCD4 at nt233-349 of 3′-UTR region. PDCD4 has been shown to inhibit the eukaryotic translation initiation factor eIF4A leading to the inhibition of procaspase-3 and p53 translation in cancer cell lines [99–102]. Research also revealed that microRNA-21 targets TPM-1, a cytoskeletal and apoptotic protein. TPM-1 was found to be downregulated in cervical cancer tissues in which microRN-21 is overexpressed [103]. It is thought to prevent proper microfilaments assembly thereby increasing malignant transformation of the cells. A cell cycle regulator protein BTG-2 is another target of microRNA-21 in cervical cancer progression [85, 104, 105]. BTG-2 is involved in various biological functions in malignant cells acting as a tumor suppressor protein. In addition, microRNA-21 targets Fas ligand (FasL) which is a part of the TNF family and activates apoptosis by binding to its cell surface receptor (FASR). The binding of the ligand induces apoptosis by the death-inducing signaling complex that includes FADD. In cancerous cells overexpressed microRNA-21 down-regulates FasL and thus its pro-apoptotic signaling leading to apoptosis inhibition [106].

Furthermore, to induce cancer proliferation, migration, invasion, and apoptosis inhibition, mircoRNA-21 directly targets GAS-5 and VHL pathways. Being a tumor suppressor, GAS-5 is downregulated both in cervical cancer cell lines and cervical cancer tissue. A luciferase reporter assay has also ascertained that there is a reciprocal repression of gene expression between microRNA-21 and GAS5, and are targets of each other. GAS5 inhibits the expression of microRNA-21. Conversely, microRNA-21 represses GAS5 expression [39]. GAS5 bring about its effects through p53 network, mTOR, and AKT signaling pathways [107]. Besides, VHL is another target of microRNA-21 to lead to proliferation, invasion, migration and inhibition of apoptosis. The 3′-UTR of von Hippel-Lindau tumor suppressor (VHL) mRNA sequence is a direct target of microRNA-21 [108].

Finally, in a study, it was proved and confirmed that microRNA-21 has a putative binding site within the 3'UTR of CCL20 using luciferase activity assays and bioinformatic analysis. The study also indicated that microRNA-21 could change proliferation, migration and apoptosis possible by controlling CCL20 in human cervical cancer cells, but the mechanism was not made clear [27].

## Meta-analysis

**Diagnostic value of microRNA-21 on cervical cancer.** Sensitivity analysis was conducted in the 8 reports of 7 included studies and two AUC reports from one study [74] were found to be outliers (S1 Fig). Therefore, the two reports have been dropped in the meta-analysis of AUC of studies although no significant change has been shown in the interpretation of the result. The pooled estimate of the effect size without dropping the two reports was 0.86 (95% CI: 0.81,0.91). As a result, the meta-analysis of the six eligible studies demonstrated that the overall area under the curve (AUC) of summary receiver operating characteristic (SROC) of microRNA-21 as a diagnostic accuracy index for cervical cancer was 0.80 (95% CI: 0.75, 0.86) (Fig 4). In addition, the pooled estimates of the other diagnostic parameters for microR-21 in

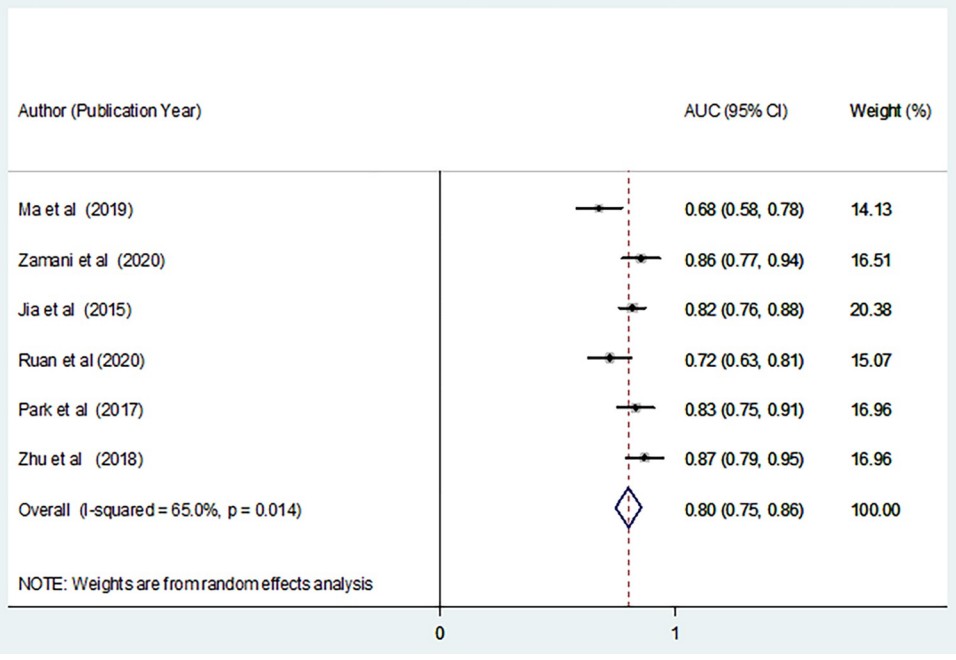

**Fig 4. A forest plot depicting the area under the curve (AUC) of summary receiver operating characteristics (SROC) of microRNA-21 assay diagnostic value in cervical cancer.**

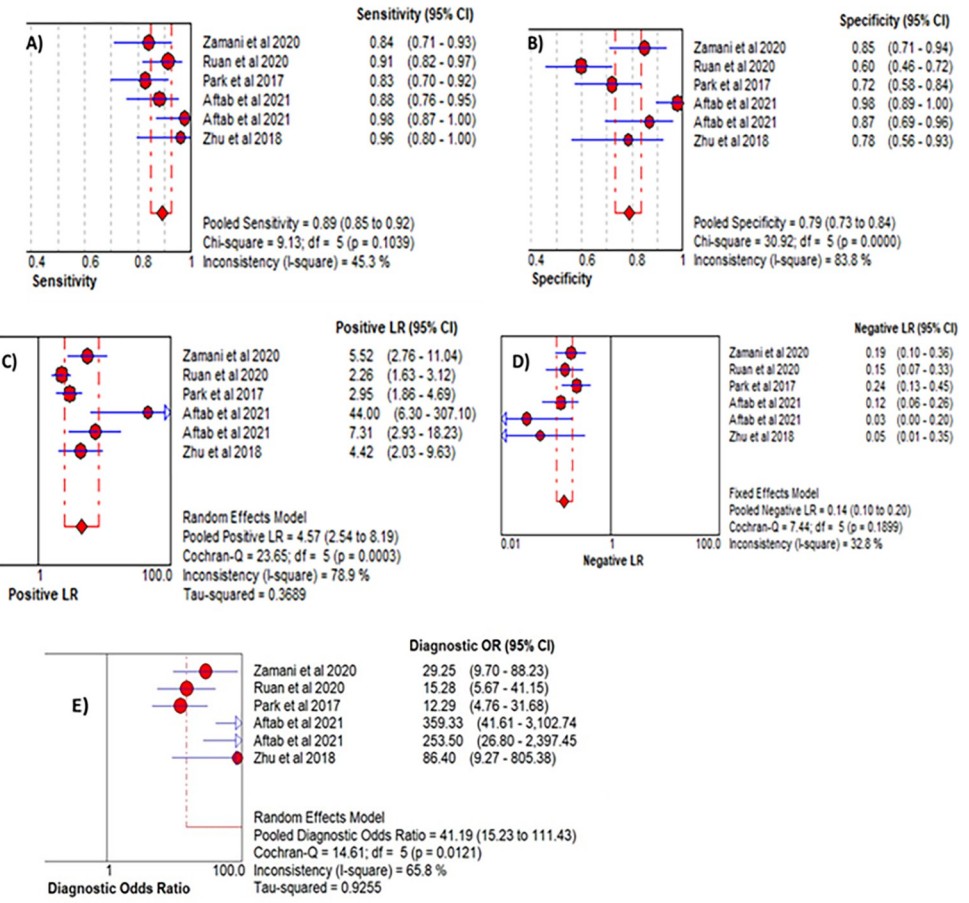

**Fig 5. Forest plots depicting the pooled sensitivity, specificity, PLR, NLR, and DOR of microRNA-21 assay diagnostic value in cervical cancer (LR: Likelihood ratio; OR: odds ratio; CI: confidence interval).**

cervical cancer were: sensitivity, 0.89 (95% CI: 0.85, 0.92); specificity, 0.79 (95% CI: 0.73, 0.84); positive likelihood ratio (PLR), 4.57 (95% CI: 2.54, 8.19); negative likelihood ratio (NLR), 0.14 (95% CI: 0.10, 0.20); and diagnostic odds ratio (DOR), 41.19 (95% CI: 15.23, 111.43) (**Fig 5A–5E**).

As it has been depicted in **Figs 4** and **5**, significant heterogeneity ($I^2 > 50\%$, $p < 0.10$) has been detected in the studies for pooling AUC, specificity, PLR, and DOR. As a result, the random-effects model has been used for the analyses. On the other hand, the fixed effects model was carried out for pooling sensitivity and NLR since no significant interstudy inconsistency was observed among the eligible studies ($I^2 < 50\%$, $p > 0.10$). A statistically significant publication bias ($p < 0.05$) was also detected by Egger's and Begg's tests. A funnel plot has also depicted that the dots, representing the studies, were not symmetrical indicating the presence of publication bias (**Fig 6**). As a result, Trim and Fill analysis was carried out to adjust the final pooled estimates.

In this meta-analysis, a sub-group analysis has also been carried out based on the sample source in the included studies. As a result, the pooled AUC of reports from tissue biopsy samples is 0.85% (95% CI: 0.81, 0.90) whereas it is 0.75 (95% CI: 0.66, 0.84) in other body fluid sources (**Fig 7**).

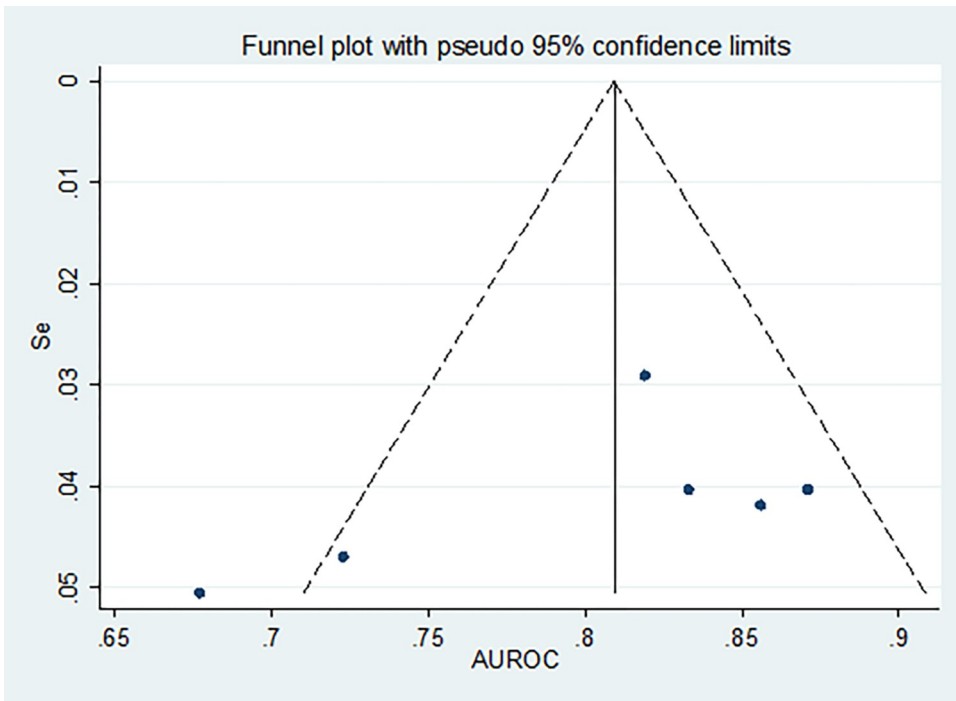

**Fig 6. A funnel plot showing the publication bias.**

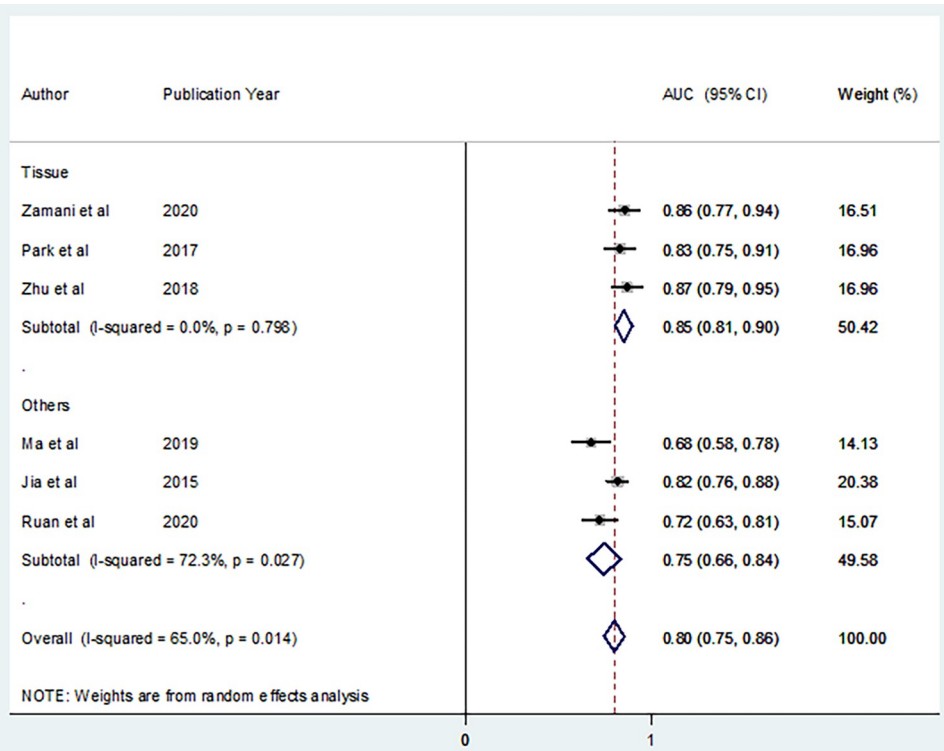

**Fig 7. A forest plot showing subgroup analysis based on the sample source of microRNA-21 assay diagnostic value in cervical cancer.**

## Prognostic value of microrna-21 on cervical cancer

Data from studies revealed that upregulated microRNA-21 led to worsening progression and poor prognosis in cervical cancer patients [45, 77]. A study has shown it to be an independent prognostic marker of the clinical outcomes of cervical cancer patients [45]. In that study, vascular invasion, depth of invasion, and lymph node metastasis were demonstrated to be significantly associated with high microRNA-21 expression (with p< 0.05 for each). In addition, another study conducted involving 112 cases and 90 healthy controls revealed that patients with high serum microRNA-21 expression had poorer recurrence free survival than patients with low microRNA-21 expression (P <0.05) [44]. Also, the study specifically revealed that high serum microRNA-21 is closely associated with lymph node metastasis in cervical cancer patients. On the other hand, a study revealed, in univariate Cox proportional hazards regression analysis, that there was a significant association between high microRNA-21 expression and poor overall survival of cervical cancer patients (P <0.05). However, the study revealed insignificant association in the multivariate Cox regression analysis (P <0.05) which was analyzed within the context of many clinical characteristics, including HPV infection, age at diagnosis, parity, clinical stage, marriage age, and histopathological grade [74]. Besides, a study conducted in China showed that the 5-year survival rate of patients with highly expressed microRNA-21 was lower than those of patients with lowly expressed microRNA-21 (p<0.05) [109]. What is more, another study revealed, in a univariate cox regression analysis, that microRNA-21 was significantly associated with cervical cancer survival [110].

## Discussion

Many studies have most consistently revealed the upregulated expression of microRNA-21 in cervical carcinogenesis process [41, 73, 98, 99, 111–113]. It has quite a complex role in cancer development and tumorigenesis with many targets in which a variety of mechanisms play a role. Relevant articles carried out on the concern were, therefore, identified so as to summarize the findings, and investigate the premise of suitability and usability of microR-21 for the diagnosis and prognosis of cervical cancer.

This study summarized that microRNA-21 targets the expression of numerous genes that regulate their subsequent downstream signaling pathways. The targets addressed in this study included TNF-α, CCL20, PTEN RasA1, TIMP3, PDCD-4, TPM-1, FASL, BTG-2, GAS-5, and VHL. Through these targets, upregulated microRNA-21 modulates a variety of signaling pathways in cervical cancer with overall effect of increased angiogenesis, uncontrolled cell cycle and cell proliferation, migration, invasion, metastasis, immune response, and inhibited apoptosis. It has become evident that HPV infection alone is not a cause for malignant transformation of the cervical cells and the alterations in the functions and synthesis of other genes take part in the pathological process of cervical tumor [114]. A recent related review showed that the evolutionarily conserved microRNA-21 is upregulated in various solid and hematological malignancies and is affiliated with high cell proliferation, high invasion, anti-apoptosis, and metastatic potential by targeting the expression of several genes [115]. It is a research hotspot that microRNAs are associated with the development, progression, invasion, metastasis, and apoptosis of cancers, acting as an oncogene or a tumor suppressor gene. [116].

Inveterate evidence on upregulated expression of microRNA-21 in cervical cancer cells and the identification of its target signaling pathways pinpoints using microRNA-21 as a convenient diagnostic and prognostic biomarker of cervical cancer. Changes in the levels of expression of miRNAs can stably and easily be assayed in tumor tissues, plasma, serum, and urine samples [74, 85]. As a result, microRNAs are potential cancer biomarkers. In this study, relevant studies have exhaustively been identified to meta-analyze the diagnostic importance of

microRNA-21 in cervical cancer. The overall pooled estimate from the meta-analysis revealed that microRNA-21 has an AUC of 0.80 with a sensitivity and specificity of 89% and 79%, respectively, for the diagnosis of cervical cancer. This finding suggests that the index test has good diagnostic accuracy for the diagnosis of cervical cancer. A recently conducted meta-analysis on various cancer types has indicated AUC to be 0.86, similar with the present meta-analysis, with other comparable diagnostic indices [117]. Another updated meta-analysis conducted on colorectal cancer also indicated the AUC to be about 0.87 [118]. Studies have shown that microRNA-21 is highly overexpressed in cervical cancer and it is involved in cell proliferation and invasion by targeting different signaling pathways [76–80]. Therefore, it can be accentuated that one target and novel stratagem to curtail cellular proliferation in preventing and treating cervical cancer could be regulating microRNA-21 expression.

In addition, this study summarized that upregulated microRNA-21 led to worsening progression and poor prognosis in cervical cancer patients. Concordant to this study, recent meta-analyses conducted in glioma and squamous cell carcinoma patients indicated microRNA-21 to be a significant and independent prognostic biomarker for survival of the patients [119, 120]. Another recent study also identified microRNA-21 to have prognostic value for breast cancer patients' survival [121]. Patients with high expression of microRNA-21 had poor outcome in terms of survival rate. This role comes by virtue of oncogenic nature of microRNA-21 that regulates many downstream effectors associated with hematological and solid malignancies [122].

## Limitation of the study

In the eligible studies, cut-off values for the expression of microRNA-21 as high and low were not uniform. In addition, because of the dearth of sufficient published research with necessary data to be extracted, a statistical summary estimate of prognostic role of microRNA-21 for cervical cancer was not pooled, and the findings of eligible studies were only qualitatively summarized.

## Conclusion

From the existing evidence, it can firmly be concluded that microRNA-21 is an oncogenic miRNA molecule playing a key role in the oncogenic phenotype of development and progression of cervical malignancy. It has a good diagnostic accuracy in the diagnosis of cervical cancer. In addition, the upregulation of microRNA-21 predicts a worse outcome in terms of prognosis in cervical cancer patients. Therefore, it can be recommended that targeting microRNA-21 and its bona fide downstream signaling targets could be tapped to develop therapeutic interventions.

## Supporting information

**S1 Table. PRISMA 2020 checklist.** The updated protocol was followed as a guideline.
(DOCX)

**S2 Table. Articles searching history from different databases.** Studies were searched from databases using key terms.
(DOCX)

**S3 Table. Quality evaluation result of disease progression and prognostic component of this study.** New Castle Ottawa Scale was used for prognostic component.
(DOCX)

**S1 Fig. Supporting file for dropping the outlier reports by two steps.** An output from sensitivity analysis.
(TIF)

## Acknowledgments

Getachew Mulu (GM), assistant professor in Debre Markos University, Ethiopia is acknowledged for supporting us in the review particularly article search process. In addition, Mekonnen Sisay (MS), assistant professor of Pharmacology at Haramaya University, Ethiopia, is also acknowledged for helping in the quality assessment and data extraction process.

## Author Contributions

**Conceptualization:** Alemu Gebrie.

**Data curation:** Alemu Gebrie.

**Formal analysis:** Alemu Gebrie.

**Methodology:** Alemu Gebrie.

**Software:** Alemu Gebrie.

**Validation:** Alemu Gebrie.

**Writing – original draft:** Alemu Gebrie.

**Writing – review & editing:** Alemu Gebrie.

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
