## [Decision Letter · Decision Letter 0]

30 Mar 2022

PONE-D-22-03455Disease progression role as well as the diagnostic and prognostic value of MicroRNA-21 in patients with cervical cancer:  A Systematic Review and Meta-AnalysisPLOS ONE

Dear Dr. Gebrie,

Thank you for submitting your manuscript to PLOS ONE. After careful consideration, we feel that it has merit but does not fully meet PLOS ONE’s publication criteria as it currently stands. Therefore, we invite you to submit a revised version of the manuscript that addresses the points raised during the review process.

 Please address the comments raised by both the reviewers for further improvement of the manuscript.

We look forward to receiving your revised manuscript.

Kind regards,

Khushboo Irshad, Ph.D.

Academic Editor

PLOS ONE

Journal Requirements:

-https://obgyn.onlinelibrary.wiley.com/doi/10.1002/ijgo.13879

- https://apps.who.int/iris/bitstream/handle/10665/349093/9789240038462-eng.pdf?isAllowed=y&sequence=1

In your revision ensure you cite all your sources (including your own works), and quote or rephrase any duplicated text outside the methods section. Further consideration is dependent on these concerns being addressed.

Reviewers' comments:

Reviewer's Responses to Questions

**Comments to the Author**

1. Is the manuscript technically sound, and do the data support the conclusions?

Reviewer #1: Partly

Reviewer #2: Yes

2. Has the statistical analysis been performed appropriately and rigorously? 

Reviewer #1: No

Reviewer #2: Yes

3. Have the authors made all data underlying the findings in their manuscript fully available?

Reviewer #1: Yes

Reviewer #2: Yes

4. Is the manuscript presented in an intelligible fashion and written in standard English?

Reviewer #1: Yes

Reviewer #2: Yes

5. Review Comments to the Author

Reviewer #1: 1) RT-PCR

Almost all the articles included in the meta-analysis had mirna-21 assayed with RT-PCR. Why weren’t efforts made to include miRNA studies were the value of mirna-21 assayed via other platforms such as sequencing. I believe adding such studies would not increase the sample size of the study but will also help categorizing a consistent effect of mirna-21 in predicting disease status but also across other assays.

2) Heterogeneity with AUC

It is not surprising to see the heterogeneity for AUC values across the included studies. The values of mirna-21 will highly deviates based on clinical predictors as exposure to HPV, age at diagnosis, family history, and other comorbidities, etc. The authors should explicitly summarize these patient characteristics.

In all the presented studies was mirna-21 the only predictor used? Did any study use any other mirna along with mirna-21 for the analysis?

3) Outliers

A study reported by Aftab et.al has shown two AUC’s > 95%. This study possibly has a class bias, and the data is skewed towards one class. It will be interesting to see if the analysis is repeated by dropping these two studies and the meta-analysis is redone.

4) Miscellaneous

a) Please change microR-21 to microRNA-21 at all instances required.

b) I recommend including more articles by expanding the publication dates. The current window of ~2 months is very limited to perform a metanalysis.

c) Would it be possible to include studies that has not reported the AUC? The authors can include effect size as a metric in the analysis and add a separate analysis.

Reviewer #2: Topic is interesting and study is good. There are few revisions suggested to the authors.

1. Results about qualitative analysis (n =40, disease progression); (n=6, prognosis) needs to be evident for this a table for charesteritics of included studies may be given as a supplementary table.

2.Results for Egger's test and Begg's test not given.

3. Justification for using various different statisticak packages for metaanalyis (MedCalc- version 20.023, Review Manager 5.3, and Stata 11.0)

4 Abbreviation AUROC and AUC define when appeared first time in text of paper.

5.Though it is given that data extraction done by two authors but no mention about literature search done by whom/ How many authors independently.

6.Method section please rewrite the scentence"The studies have been retrieved from November 26, 2021 to January 18, 2022 and all the 162 articles accessed until January 18, 2022" it is confusing and appears that these are the cut off dates for literature search period.

7.Table 2 not clear : in prisma chart no. of included studies is given 7 in table it is given as 10 out of which two studies with * mark excluded so number comes is 9 . Please cross check no of studies include in metaanalyis.

8. Table 2 Again this table is regarding the characteristics of included studies in the systematic review and meta-analysis and authors included two studies that are not included in metaanalyis then why these studies included in table 2 . Either delete or give the justification in foot note. Expansion of AUROC also needs to be given in foot note.

9. In figure 4 for metaanalysis of sensitivity and specificity what model used (fixed or random ) not give. In foot note expand the abbriviations used in figure 4.

10 how the quality assessment done for studies included in qualitative analysis.

11. Contribution of each author needs to be given and not just one author.

6. PLOS authors have the option to publish the peer review history of their article (what does this mean?). If published, this will include your full peer review and any attached files.

Reviewer #1: **Yes: **Hrishikesh Lokhande

Reviewer #2: No

---

## [Author Response · Author response to Decision Letter 0]

18 Apr 2022

Dear Editors!

PLOS ONE 

 A letter Accompanying Revision in Response to Reviewers’ Comments

I am pleased to resubmit for publication version of “Disease progression role as well as the diagnostic and prognostic value of MicroRNA-21 in patients with cervical cancer: A Systematic Review and Meta-Analysis” for a review as original research in PLOS ONE.

The comments of the reviewers were highly insightful and enabled me to greatly improve the quality of the manuscript. Therefore, based on the reviewers’ concerns, I have made extensive revision in the manuscript. In the following pages, I have addressed yours’ concerns in a point by point format. 

I look forward to hearing from you at your earliest convenience. 

Thank you for your consideration of this manuscript! 

Sincerely, 

Alemu Gebrie 

Response to reviewers' comments

Reviewer #1: 

1. RT-PCR

Almost all the articles included in the meta-analysis had mirna-21 assayed with RT-PCR. Why weren’t efforts made to include miRNA studies were the value of mirna-21 assayed via other platforms such as sequencing. I believe adding such studies would not increase the sample size of the study but will also help categorizing a consistent effect of mirna-21 in predicting disease status but also across other assays.

Response: Thank you for raising an important issue. It is right that microRNA molecules are assayed using different methods. No restriction has been put to an assay method of micro-RNA-21 in this meta-analysis study. Unfortunately, all the 7 studies included in the meta-analysis used RT-PCR as an assay method. If there were an article with other assay mothods such as sequencing and microarray on microRNA-21, it would not be excluded in the study. Even, we would do a subgroup analysis using the assay method as a factor.

2. Heterogeneity with AUC

It is not surprising to see the heterogeneity for AUC values across the included studies. The values of mirna-21 will highly deviates based on clinical predictors as exposure to HPV, age at diagnosis, family history, and other comorbidities, etc. The authors should explicitly summarize these patient characteristics.

In all the presented studies was mirna-21 the only predictor used? Did any study use any other mirna along with mirna-21 for the analysis?

Response: Thank you for the important comment and issues. In Table 2 some study subjects’ characteristics are presented despite the incomplete report in the studies. The problem is studies do not report all those characteristics to extract the data to this study. microRNA-21 is not the only microRNA related with cervical cancer. There are microRNA molecules up or down regulated in cervical cancer. In addition, there are studies that address microRNA molecules both single and as a signature of panels of microRNA molecules. However, the objective of this study is to review the disease progression role as well as the diagnostic and prognostic role of microRNA-21 in cervical cancer. This microRNA is chosen because of the fact that it is the most constituvely overexpressed one in cancer. 

3. Outliers

A study reported by Aftab et.al has shown two AUC’s > 95%. This study possibly has a class bias, and the data is skewed towards one class. It will be interesting to see if the analysis is repeated by dropping these two studies and the meta-analysis is redone.

Response:Thank you very much for that critical suggestion. Of course, the sensitivity analysis showed that the two reports from the study were extreme values. In the previous manuscript the two studies were not dropped by considering a couple of reasons that the study has no quality problem, the methodology of the study was ok and random effects model was used for adjusting the weight taken by the study on the pooled AUC. In addition, the interpretation of the result is not well changed without dropping the study. However, I must accept that critical suggestion that it is recommened to drop an outlier study from a meta-analysis sothat the result will be more acceptable and credible. Therefore, the manuscript has properly been revised that the two reports are now dropped and almost all statistical analyses are redone. In addition, a supporting file 4 (as a figure) indicating the result of sensitivity analysis has been attached. 

4. Miscellaneous

a) Please change microR-21 to microRNA-21 at all instances required.

 Response: Thank you. This has been addressed. Kindly, see the track change.

b) I recommend including more articles by expanding the publication dates. The current window of ~2 months is very limited to perform a metanalysis.

 Response: Thank you. As it is indicated in the methods section, the articles have been exhaustively searched from several databases using various key terms for about two months. Articles were searched very exhaustively until saturation. In addition, no publication date restriction for an article was made in searching the articles in the meta-analysis before or after our searching period. If there is any relevant study found not included in the analysis (found at any stage until publication), the analysis can be redone and updated. However, there is, of course, a time period for searching, data extraction, analysis, write up and journal processing for publication after submission. 

c) Would it be possible to include studies that has not reported the AUC? The authors can include effect size as a metric in the analysis and add a separate analysis.

 Response: Thank you for the concern. The effect size is not limited to AUC but it includes other effect sizes as well such as sensitivity, sepecificity and others listed in the meta-analysis. However, if studies report sensitivity and specificity, it is possible to calculate the AUC. That is why the studies included in the metaanalysis of AUC are 7 in number (8 reports) and the reports are only 6 in number in the case of meta-analysis of other effect sizes as shown in Fig 4 and 5. If sensitivity and specificity are given then it is possible to calculate other diagnostic indices in the original studies during data extraction. 

Reviewer #2: 

Topic is interesting and study is good. There are few revisions suggested to the authors.

1. Results about qualitative analysis (n =40, disease progression); (n=6, prognosis) needs to be evident for this a table for charesteritics of included studies may be given as a supplementary table.

Response: Thank you for that important comment. The manuscript has been based on the comment and (S3 Tble) has been added a supplementary table.

2. Results for Egger's test and Begg's test not given.

Response: Thank you for the question. However, in the results section paragraph 2, It is stated that a statistically significant publication bias (p <0.05) was detected by Egger’s and Begg’s tests. Both the objective and subjective analysis for publication bias were conducted and a significant publication bias was detected and a trim fill analysis has been carried out as a measurement. Also, the tests are presented in figure 6 (see the track change)

3. Justification for using various different statisticak packages for metaanalyis (MedCalc- version 20.023, Review Manager 5.3, and Stata 11.0)

Response: Thank you for the relevant question that needs clarification and and justification. Different statistical analyses were carried out in this meta-analysis. An analysis which was not suitably conducted in one software was conducted by the other software. RevMan 5.3 software was used to do the analysis of quality of studies using QUADAS-2 tool. The meta-analysis of AUC, funnel plot, egger’s test were carried out using stata software. In addition, as indicated in figure 5. The statistical analysis for sensitivity, specificity, DOR etc, the possible software to pool data was MetaDisk, because the effect sizes can be pooled from true positive, false positive, false negative, true negative values. But, there are studies which presented AUC only without these values. Finally, medcalc is used to convert from true positive, false positive, false negative, true negative values to AUC. Generally, the data presented in the original studies (some studies reported AUC only even the point estimate only, others sensitivity and specificity and some true positive-true negative values) for the meta-analysis are reported in different ways and to manipulate and analyse in a more meaningful form, different software were used. 

4. Abbreviation AUROC and AUC define when appeared first time in text of paper.

Response: Thank you. This has been addressed. Kindly, see the track change.

5. Though it is given that data extraction done by two authors but no mention about literature search done by whom/ How many authors independently.

Response: Thank you for raising the concern. This has been addressed in the acknowledgement section and in the methods section. 

6. Method section please rewrite the scentence"The studies have been retrieved from November 26, 2021 to January 18, 2022 and all the 162 articles accessed until January 18, 2022" it is confusing and appears that these are the cut off dates for literature search period.

Response: Thank you for raising the issue. As it is indicated in S2 Table (Supporting file 2), exhaustive search of potential articles has been conducted in about 7 databases for about two months from November 26, 2021 to January 18, 2022. It is the literature searching period from the databases.

7. Table 2 not clear : in prisma chart no. of included studies is given 7 in table it is given as 10 out of which two studies with * mark excluded so number comes is 9 . Please cross check no of studies include in metaanalyis.

Response: Thank you for the concern. I see the concern. As it is clearly stated in the “Description of the included Studies” subtitle, a total of 9 studies with 10 reports of the diagnostic importance of microRNA-21 in cervical cancer were included in this study, the two studies being not included in the meta-analysis. Table 2 contains 9 studies with one study (Aftab et al) having two reports. That means, one study is presented twice in the table because of its two reports. Therefore, 9 sudies with a total of 10 reportes are indicated in the table and 7 studies with a total of 8 reports are included in the meta-analysis of AUC effect size. For more clarity, the title of Table 2 has been modified. Kindly see the track change.

8. Table 2 Again this table is regarding the characteristics of included studies in the systematic review and meta-analysis and authors included two studies that are not included in metaanalyis then why these studies included in table 2 . Either delete or give the justification in foot note. Expansion of AUROC also needs to be given in foot note.

Response: Thank you for the constructive concern. This has been addressed. Kindly, see the track change. A justification has been given that the two studies cannot be included in the meta-analysis because only point estimates can be extracted from the articles. Without the respective confidence intervals or standard errors it is not possible to pool data. However, it is good to, at least, present and describe the related data in the table. 

9. In figure 4 for metaanalysis of sensitivity and specificity what model used (fixed or random ) not give. In foot note expand the abbriviations used in figure 4.

Response: Thank you. The abbriviations have been expanded in the foot note. Kindly, see the track change. Although not displayed in the figure (the software output did not show the model), the models used for sensitivity and specificity has been stated in the text describing the table that significant heterogeneity (I2>50%, p<0.10) has been detected in the studies for pooling AUC, specificity, PLR, and DOR. As a result, the random-effects model has been used for the analyses. On the other hand, the fixed effects model was carried out for pooling sensitivity and NLR since no significant interstudy inconsistency was observed among the eligible studies (I2<50%, p>0.10). 

10. how the quality assessment done for studies included in qualitative analysis.

Response: Thank you for the constructive comment. Quality assessment was only addressed for the meta-analysis part in the previous version of the manuscript. The current version of the manuscript, however, has been revised that new castle quality assessment was done for the studies included in the prognostic component. In addition, the journal ranking quality of journals of the included studies in the disease progression role component has been indicated in S3 table as a supplemetntary table.

11. Contribution of each author needs to be given and not just one author.

Response: Thank you for the comment. While the contribution of the author of this study is stated clearly in the section, the other two reviewers/persons who supported the author have been acknowledged in the acknowledgement section for their contribution/help.

Additional requirements 

Response: The manuscript has been revised based on the comment.

-https://obgyn.onlinelibrary.wiley.com/doi/10.1002/ijgo.13879

- https://apps.who.int/iris/bitstream/handle/10665/349093/9789240038462-eng.pdf?isAllowed=y&sequence=1

Response: Thank you for the concern. The publication is quite different from the present study. The texts of this study are paraphrased. If there is any specific overlap, is can be revised.

---

## [Decision Letter · Decision Letter 1]

1 May 2022

Disease progression role as well as the diagnostic and prognostic value of MicroRNA-21 in patients with cervical cancer:  A Systematic Review and Meta-Analysis

PONE-D-22-03455R1

Dear Dr. Alemu Gebrie,

We’re pleased to inform you that your manuscript has been judged scientifically suitable for publication and will be formally accepted for publication once it meets all outstanding technical requirements.

Kind regards,

Khushboo Irshad, Ph.D.

Academic Editor

PLOS ONE

Reviewers' comments:

Reviewer's Responses to Questions

**Comments to the Author**

1. If the authors have adequately addressed your comments raised in a previous round of review and you feel that this manuscript is now acceptable for publication, you may indicate that here to bypass the “Comments to the Author” section, enter your conflict of interest statement in the “Confidential to Editor” section, and submit your "Accept" recommendation.

Reviewer #2: All comments have been addressed

2. Is the manuscript technically sound, and do the data support the conclusions?

Reviewer #2: Yes

3. Has the statistical analysis been performed appropriately and rigorously? 

Reviewer #2: Yes

4. Have the authors made all data underlying the findings in their manuscript fully available?

Reviewer #2: Yes

5. Is the manuscript presented in an intelligible fashion and written in standard English?

Reviewer #2: Yes

6. Review Comments to the Author

Reviewer #2: All the issues are addressed successfully, only a minor comment/suggestion to give individual authors contributions as acknowledgement is to acknowledge work of others and not the authors.

7. PLOS authors have the option to publish the peer review history of their article (what does this mean?). If published, this will include your full peer review and any attached files.

Reviewer #2: No

---

## [Editor Report · Acceptance letter]

1 Jul 2022

PONE-D-22-03455R1 

Disease progression role as well as the diagnostic and prognostic value of microRNA-21 in patients with cervical cancer:  a systematic review and meta-analysis 

Dear Dr. Gebrie:

I'm pleased to inform you that your manuscript has been deemed suitable for publication in PLOS ONE. Congratulations! Your manuscript is now with our production department. 

Kind regards, 

on behalf of

Dr. Khushboo Irshad 

Academic Editor

PLOS ONE